# Current Perioperative Care in Pancreatoduodenectomy: A Step-by-Step Surgical Roadmap from First Visit to Discharge

**DOI:** 10.3390/cancers15092499

**Published:** 2023-04-26

**Authors:** Tommaso Giuliani, Giampaolo Perri, Ravinder Kang, Giovanni Marchegiani

**Affiliations:** 1Memorial Sloan Kettering Cancer Center, New York, NY 10065, USA; giuliat@mskcc.org (T.G.); kangr1@mskcc.org (R.K.); 2Verona University Hospital, 37134 Verona, Italy; giampaolo.perri@univr.it; 3Department of Surgical, Oncological and Gastroenterological Sciences (DiSCOG), University of Padua, 35122 Padua, Italy

**Keywords:** pancreatoduodenectomy, preoperative risk, frailty, mitigation strategies, enhanced recovery after surgery, pancreatic fistula

## Abstract

**Simple Summary:**

The treatment of periampullary tumors is becoming increasingly multimodal. Surgical interventions must be weighed against alternative therapeutic options, including ablative radiation and systemic chemotherapy. An operation should not jeopardize the receipt of adjuvant therapies. Therefore, a multiparametric risk assessment is crucial for patients who are potential surgical candidates. If pancreatoduodenectomy (PD) is feasible, a series of perioperative checkpoints are keys to a rapid recovery. The aim of our review is to itemize the pre-, intra-, and post-operative pathways of PD, with the intent of providing clinicians with an up-to-date perioperative roadmap.

**Abstract:**

Pancreaticoduodenectomy (PD) is a mainstay in the management of periampullary tumors. Treatment algorithms increasingly employ a multimodal strategy, which includes neoadjuvant and adjuvant therapies. However, the successful treatment of a patient is contingent on the execution of a complex operation, whereby minimizing postoperative complications and optimizing a fast and complete recovery are crucial to the overall success. In this setting, risk reduction and benchmarking the quality of care are essential frameworks through which modern perioperative PD care must be delivered. The postoperative course is primarily influenced by pancreatic fistulas, but other patient- and hospital-associated factors, such as frailty and the ability to rescue from complications, also affect the outcomes. A comprehensive understanding of the factors influencing surgical outcomes allows the clinician to risk stratify the patient, thereby facilitating a frank discussion of the morbidity and mortality of PD. Further, such an understanding allows the clinician to practice based on the most up-to-date evidence. This review intends to provide clinicians with a roadmap to the perioperative PD pathway. We review key considerations in the pre-, intra-, and post-operative periods.

## 1. Introduction

The importance of perioperative risk assessment cannot be overstated. Pancreaticoduodenectomy (PD) is associated with morbidity as high as 60%, and mortality approaching 10%, at low volume centers [1,2,3]. Thus, the decision to proceed with PD must be weighed against the following considerations:Are non-surgical options available?

Radiation therapy (RT) can be used to palliate symptoms in unresectable pancreatic cancer. In addition, it may also serve as an alternative to surgery in selecting patients with locally advanced disease. In a recent retrospective study from Memorial Sloan Kettering Cancer Center, patients with locally advanced pancreatic adenocarcinoma received induction chemotherapy followed by either ablative RT or PD, and it was noted that the cumulative incidence of local recurrence was similar between patients undergoing PD and those receiving definitive ablative RT. However, as expected, the distant progression and overall survival (OS) were better in the surgical group. Yet, the OS was more than 20 months for patients receiving ablative RT, which is higher than historic cohorts that received surgery alone [4,5]. In the past, nonsurgical treatment options provided marginal survival benefits; however, this study demonstrates that RT may be a viable alternative to surgery in selected patients [6].

2.Would PD prevent the receipt of chemotherapy?

The importance of adjuvant chemotherapy was established in the early 2000s, when Neoptolemos and colleagues demonstrated a survival benefit among patients who received adjuvant chemotherapy [7]. Newer drug regimens demonstrate a median overall survival of as long as 54 months, compared to 20 months in historical surgery-only cohorts [8,9]. Thus, pancreatic resection should not jeopardize the initiation of adjuvant treatment. Yet, a nationwide Dutch study demonstrated that a third of patients failed to receive chemotherapy after resection of pancreatic ductal adenocarcinoma (PDAC). The authors noted advanced age, poor performance status, and surgical complications as the main factors that limited the initiation of adjuvant therapy [10].

3.Can surgery be safely deferred?

One indication for PD is intraductal papillary mucinous neoplasms (IPMN) with high-risk features. However, a meta-analysis demonstrated that PD can be deferred in patients with IPMN and worrisome features or high-risk stigmata. These patients were deemed unfit for surgery, but surveillance was continued. The study investigators found that IPMN-related mortality is low—in fact, lower than the risk of death from other causes [11]. This study suggests that a conservative/non-operative approach can be justified in selecting patients with IPMN, even when the pancreatic cysts have high-risk features. Ultimately, if the patient has severe comorbidities, their overall life expectancy and prognosis are not impacted by the risk of developing a PDAC.

After these questions are properly addressed and the patients are scheduled for surgical resection, treatment success is contingent on the execution of a complex operation, such as PD, where minimizing postoperative complications and optimizing a fast and complete recovery are crucial. Risk reduction and benchmarking are essential frameworks through which modern perioperative care must be delivered in the setting of PD. While the postoperative course is primarily influenced by pancreatic fistulas (POPFs), many other patient- and hospital-associated factors, such as frailty and the ability to rescue from complications, also affect the outcomes. A comprehensive understanding of these factors allows the patient’s risk stratification and facilitates a frank discussion of the morbidity and mortality of PD. Further, such an understanding allows the clinician to practice based on the most up-to-date evidence. This review intends to provide clinicians with a roadmap to the perioperative PD pathway in the pre-, intra-, and post-operative periods (Figure 1).

## 2. Preoperative Period

### 2.1. Age

The patient’s age is one of the first objective data points gathered during the initial consultation. Temporal trends of PD note that the average age of surgical patients has increased and will continue to do so in the coming years [12]. The impact of age on outcomes is complex. In one single center retrospective study, patients older than 75 years of age were compared to two matched cohorts (ages 65–75 and 40–65). There was no statistically significant difference in post-operative outcomes (complications or mortality) by age [13]. Yet, another multicenter retrospective case–control study found that 90-day mortality following PD in octogenarian was 9% compared to 3% in the younger controls [14]. Interestingly, this study found comparable rates of postoperative complications between the octogenarians and younger patient cohort. The difference in mortality may be due to a decreased functional reserve in the octogenarian group or may potentially represent a failure to rescue (FTR). In a large dataset of 22,983 patients who underwent PD at European and American centers, age was a preoperative predictor of FTR both on univariate and multivariate analysis [15]. An American Society of Anesthesiology (ASA) score ≥ 3 was also associated with FTR. We will discuss the concept of frailty later in this section. For now, suffice it to say that PD will increasingly be offered to older patients. Rescuing these patients from complications can be a real challenge and postoperative mortality may be higher in this older patient population. However, definitive numeric age cut-offs cannot be drawn, as functional and biological factors must be considered in a comprehensive multiparametric assessment.

### 2.2. Nutritional Status

Patients undergoing pancreatic resections often have a poor nutritional status at baseline. In fact, 21–86% of patients with PDAC have sarcopenia [16,17]. Unfortunately, the definition of sarcopenia is not universal, thus its use as a surrogate for nutritional status is challenging. Protein energy malnutrition (PEM) is an alternative surrogate. PEM encompasses multiple conditions, including cachexia, marasmus, loss of weight, and underweight. An observational study used the National Inpatient Sample (NIS) database and compared outcomes of patients undergoing PD with PEM to those without PEM and noted a 2.25-fold higher risk of in-hospital mortality among PEM patients [18]. However, the clinical application of PEM is limited, as it is an entity defined by ICD-9 billing codes, and thus it is not a discrete tool which can be used in the preoperative setting. However, cross-sectional imaging can be used to define body composition in the preoperative setting. Sandini and colleagues used CT scans to calculate the ratio between visceral fat and total abdominal skeletal muscle [19]. This ratio was the only independent predictor of major complications following PD.

While we have focused on cachexia and loss of weight, it is important to remember that obesity can also represent poor nutritional status. In an analysis on 1865 patients undergoing PD, high BMI (>30 kg/m^2^) independently predicted mortality and FTR. In a subset analysis of patients undergoing resection for nonmalignant tumors, obesity remained an independent predictor of mortality, with FTR being 5.7-fold higher in this subset. The authors concluded that clinicians should consider deferring surgery for obese patients with nonmalignant indications to optimize preoperative conditioning [20]. Ultimately, the preoperative assessment of a patient’s nutritional status is important as it predicts surgical risk and addressing this modifiable characteristic a priori can improve outcomes.

### 2.3. Accumulated Comorbidities

The fact that comorbidities impact morbidity and mortality after pancreatic resections is well known. Several multiparametric scores have been developed and validated, among these is the Charlson Comorbidity index. This index assigns weight to each comorbidity and predicts long-term mortality. The Charlson Age Comorbidity Index (CACI) incorporates age and adds one additional point for each decade of age starting at 40 years. A study from the Massachusetts General Hospital (MGH) explored factors associated with the individual likelihood of surviving beyond 1 year after PD. They demonstrated that patients with a high CACI score (≥ 6) had a <50% likelihood of being alive 1 year postoperatively [21].

In addition to comorbidities, the functional status, and the concept of frailty must be assessed as well. The original definition of frailty refers to a model for the age-related state of decreased physiologic reserves [22,23]. This “phenotypic” model used a clinical questionnaire to assess different functional aspects of the patient’s life. Since this scale was developed in a non-surgical population, the modified frailty index (mFI) should be used instead. The latter index relies on the concept of accumulating deficit and was developed for the surgical patient population [24,25]. In a retrospective analysis of the American College of Surgeons National Surgical Quality Improvement Program (NSQIP) database, 13,020 pancreatectomies were classification based on the mFI. Patients with higher mFI had increased odds of postoperative mortality. Among the items of the mFI, functional status, congestive heart failure, and impaired sensorium/encephalopathy had the highest odds ratio for mortality (3.03, 4.77, and 5.06, respectively) [26]. Thus, available scores such as Charlson Comorbidity index and mFI should be calculated early on. These tools can be used during the pre-surgical visit to provide patients with individualized risk of surgery-related death (Table 1).

### 2.4. Predicting Fistula in Clinic

Surgeons should be aware that the occurrence of a POPF is one of the main drivers of the entire postoperative course. In fact, even though most POPFs have a benign course, some result in abdominal collections, post-pancreatectomy hemorrhage (PPH), delayed gastric emptying (DGE), sepsis, multiorgan failure, and death [27]. POPF occurs in 5–30% patients after PD, and this rate has not significantly decreased over the last few decades [2]. There are preoperative tools that can predict the risk of a pancreatic leak. These tools take into consideration the gland texture and the caliber of the main pancreatic duct (MPD). In particular, the smaller the MPD, the higher the risk of fistula. Roberts and colleagues confirmed a strong association between the MPD caliber as assessed on preoperative cross-sectional imaging and POPF [28]. The authors developed and validated a score solely based on the MPD diameter and the BMI, which has a good predictive performance.

However, a reliable estimation of gland texture in the preoperative setting is more complex. Cross-sectional imaging has been used to this end as well. Gnanasecaran and colleagues used a pancreatic enhancement ratio (PER) as a surrogate for gland texture and were able to demonstrate a moderate correlation between PER and POPF [29]. Another surrogate for gland texture is preoperative exocrine functionality, whereby normal exocrine function indicates soft parenchyma. Fecal elastase-1 levels can be used to assess exocrine function, and normal levels have been associated with an increased risk of fistula [30]. BMI can also be used as an indirect, simple yet reliable measure of gland adiposity [31]. Practical decision trees which utilize preoperative parameters such as MPD caliber and BMI have been validated and can provide the clinician with a good estimation of the future risk of fistula and prolonged hospitalization [32]. Again, these are important considerations when counseling patients preoperatively.

### 2.5. The Preoperative Biliary Drainage

Approximately half of patients with periampullary tumors present with hyperbilirubinemia at the surgical evaluation [2].

The role of preoperative biliary drainage (PBD) is still a matter of debate. On the one hand, surgery in patients with jaundice can lead to coagulopathy, hepatic and renal dysfunction, and an increased rate of postoperative complications [33]. On the other hand, PBD is a potential source of colonization of bile, which may trigger infectious complications, including sepsis [34,35]. Furthermore, PBD per se carries remarkable risks. Regardless of the approach—endoscopic (EBD) or percutaneous (PTBD)—PBD related complications occur in 2 to 23% of cases and include pancreatitis, hemorrhage, cholangitis, bile leak, and perforations [36,37].

Altogether, the risks of routine PBD overcome the theoretical benefit of jaundice palliation before surgery [38]. In a landmark RCT on this topic, patients with periampullary tumors and obstructive jaundice (bilirubin < 14.6 mg/dL) were randomized into either PBD or surgery upfront. Postoperative surgical complications and mortality were similar between the arms. However, severe complications were significantly more frequent in the PBD group when the procedure-related adverse events were considered (74% vs. 30%) [39].

The exact level of bilirubin in which palliation could be beneficial is unknow. As mentioned, an upper limit of 14 mg/dL has been suggested [39]. Selected clinical scenarios prompt a PBD regardless; these include conditions requiring an immediate decompression such as cholangitis and secondary systemic organ dysfunction, debilitating pruritus, and anytime a surgical intervention cannot be scheduled in a timely fashion [40].

### 2.6. The Volume-Outcome Relationship

Surgeons must also consider their own expertise and its impact on outcomes. Birkmeyer and colleagues demonstrated that hospital volume for complex surgical procedures, including PD-influenced perioperative outcomes. Specifically, they demonstrated that PD mortality rates varied from 16% at very low volume centers to 3.8% at high volume centers [41,42]. The volume–outcome relationship has been redemonstrated numerous times and has led many health systems to centralize pancreatic resections, often on a national level [43]. The improved outcomes at high volume centers can be partially explained by the resources of tertiary care centers, or on the cumulative knowledge of the teams involved in perioperative care. However, individual surgeons’ experience also remains an important determinant of morbidity irrespective of annual volume [44]. Interestingly, a study from the University of Michigan demonstrated that surgeons who conducted a high volume of hepato-pancreato-biliary procedures, but relatively few PDs, had outcomes similar to those who performed more PDs. This reframes the volume paradigm from volume for a specific procedure to cumulative HPB volume [45]. That being said, the thresholds to define high volume vary at both hospital and surgeon levels, but the literature generally converges to advise against performing PDs in hospitals with volumes lower than 10 cases/year [46].

## 3. Intraoperative Period

### 3.1. The Rise of Minimally Invasive Surgery

Minimally invasive surgery (MIS) has grown exponentially, with 11–30% of pancreas surgery being performed either laparoscopically or robotically [47,48]. The rise in MIS pancreas surgery has been commensurate with the trends in general surgery, as up to 15% of all general surgery procedures are currently performed robotically [49]. While a number of trials have shown that robotic surgery is safe for distal pancreatectomies, and that there is an advantage with regard to the functional recovery of patients and shorter length of stay, the same has not been proven in minimally invasive pancreatoduodenectomy [50,51,52]. To date, there are three published trials comparing MIS PD to open PD: the PLOT trial, the PADULAP trial, and LEOPARD-2. The PLOT trial was conducted in India and was a single-center trial of 64 patients who were randomized to open versus laparoscopic PD. While this study did demonstrate a shorter hospital stay (LOS) in the MIS arm, it is important to note that the primary outcome was changed during the interim analysis from complication rate to LOS [53]. A second randomized clinical trial was performed in Spain with 66 patients again showed a shorter LOS [52]. This study noted no difference in complications or oncologic outcomes. However, the PADULAP trial included patients with premalignant and benign lesions; thus, whether these findings can be extrapolated to patients with periampullary malignancies is unclear. The only multicenter trial published at this point is the LEOPARD-2 trial, in which 99 patients were randomized to MIS or open PD [54]. This trial also noted a shorter functional recovery in the MIS arm. However, the study was terminated early secondary to the Data and Safety Monitoring Board noting a higher 90-day complication rate in the MIS arm of 10% versus 2% in the open arm. Of note, all the enrolling centers included in the LEOPARD-2 trial performed at least 20 PDs annually, and all had performed 20 MIS PDs prior to trial participation. Nevertheless, these institutions were likely still on their on the learning curve. Data from University of Pittsburgh suggest that the institutional learning curve for robotic PD is at least 80 cases [55]. Thus, the implementation of laparoscopic or robotics PD must be undertaken within a formal training paradigm. Recently, the University of Pittsburgh group demonstrated that a training program which includes online videos, robotic simulations, wet-labs, and on-site proctoring could be successful applied at other institutions and, as a result, robotics can be safely introduced without a detrimental impact to patient safety [56].

### 3.2. Fistula Risk Assessment

Over the years, different prevention and mitigation strategies for patients at high-risk for POPF have been employed, and ultimately a tailored approach is favored over a “one-size-fits-all” strategy [57]. First, patients must be classified as either low- or high-risk for POPF; both pancreas-specific and patient-specific factors must be considered to make this determination. In addition, most predictive scores include intraoperative variables. The Fistula Risk Score (FRS) is undoubtedly the most widely used due to its simplicity and applicability. The FRS is based on four variables: pancreas texture, pancreatic duct size, intraoperative blood loss, and presumptive pathology. However, alternative instruments have been developed and extensively validated, such as the Alternative Fistula Risk Score (a-FRS) [31,58]. The most important pancreas-specific factors are gland texture and MPD caliber [59]. A soft gland and small MPD (<3 mm) are associated with high-risk anastomosis. While these characteristics may be interdependent, MPD and gland texture are independently associated with the risk of POPF [60]. A combination of these conditions is more often encountered in diseases that do not cause upstream obstruction of MPD (and subsequent chronic obstructive pancreatitis), or cases of diffuse adipose infiltration of the organ [61]. Consequently, duodenal, ampullary, cystic, and neuroendocrine pathologies are considered surrogate factors for POPF development, as they are rarely associated with a dilated MPD/firm texture. Conversely, PDAC and chronic pancreatitis often present with dilated MPD/fibrotic parenchyma and are therefore associated with lower risk [62]. Moreover, among PDAC patients, neoadjuvant therapy showed a relevant protective effect on POPF risk [63].

### 3.3. Fistula Mitigation Strategies

A large multicenter study investigated mitigation strategies for POPF using a risk-stratified approach (for those patients with a-FRS of 7 to 10), the optimal POPF mitigation was achieved with no prophylactic octreotide administration, pancreatojejunostomy (PJ) anastomosis with externalized trans-anastomotic stent (ETS), and the placement of abdominal drains [57].

Somatostatin analogs

The prophylactic administration of somatostatin analogs (SSA) has been extensively debated as a prevention strategy. Multiple randomized controlled trials have been conducted. A seminal trial from Memorial Sloan Kettering Cancer Center has demonstrated lower POPF with the routine use of SSA, specifically with Pasireotide^®^, disproving the former data that failed to demonstrate any benefit of SSA use [64,65]. Multiple meta-analyses have also been published on this topic, these note a lower incidence of pancreatic fistulas with SSA use; however, the proportion of fistulas that were clinically significant remains unclear [66,67]. Presently, the routine administration of prophylactic analogues of somatostatin is still controversial [57]. Of note, most of the available literature predates the preoperative risk stratification era. Thus, the selective use of SSA may be appropriate in the high-risk setting, and such a strategy would also be more cost-effective [62].

Pancreatic anastomosis technique

With regard to pancreatic anastomosis, the options for a high-risk gland are pancreaticogastrostomy (PG) or PJ. Randomized clinical trials have failed to establish the superiority of one approach over the other. A recent trial published in 2020 randomized patients to PG or PJ with external stents in the high-risk setting (FRS from 7 to 10) [68]. The rates of POPF were similar for both techniques; however, PG was associated with increased rates of major morbidity (Clavien–Dindo grade ≥3). Interestingly, a recent meta-analysis noted PG had lower rates of intraperitoneal fluid collections, but higher rates of postoperative hemorrhage, without significantly different rates of grade B/C POPF [69,70,71]

Trans-anastomotic stent

Trans-anastomotic stents can be used to mitigate the risk of POPF, but there is wide variety in the purported efficacy of such a strategy. Despite the inconsistency in the results, several randomized trials have demonstrated a significant reduction in POPF occurrence with the use of anastomotic stents [72,73,74]. A recent Cochrane review evaluating the roles of pancreatic duct stents in preventing POPF has identified eight randomized clinical trials with a total of 1018 participants, leading to the conclusion that using an anastomotic stent provides no significant differences in terms of clinically relevant outcomes [75]. However, the use of different stent types (i.e., externalized versus non-externalized), lack of a standardized technique, and lack of risk-stratified approach represent bias. An additional paper not included in the review by Jang and colleagues showed an advantage for internal over external stents, with a 5.5% reduction in POPF rate [76]. Recent evidence suggests that a trans-anastomotic stent does not reduce the POPF rate for patients in all risk categories, instead providing a real benefit only to those patients deemed at high risk of POPF [57]. Of note, ETS malfunctioning may occur in almost 20% of patients, neutralizing its beneficial effects [77,78]

Prophylactic drainage

Abdominal peri-anastomotic drains historically served two roles: (1) they allowed for monitoring of postoperative complications and (2) the management of said complication [79]. However, a wide discrepancy exists with regard to the indications for drainage use, their number, and subsequent management [80,81]. A tailored strategy should be adopted according to the risk stratification. The concept of drain omission, with the rationale of avoiding external contamination, was first proposed decades ago. However, as demonstrated by the detrimental results of one of the first trials on the topic, the indiscriminate omission of abdominal drainages after PD led to increased incidence of severe complications and higher mortality [82]. The implementation of risk stratification eventually allowed the detection of a low-risk population (FRS 0–2), in which drainage omission was safely implemented without increasing the complication burden [83]. Conversely, in both intermediate- and high-risk patients, drainage appears to be beneficial, as it allows prompt identification of complications and facilitates their management.

## 4. Postoperative Period

### 4.1. ERAS in Pancreatic Surgery

Enhanced Recovery After Surgery (ERAS) pathways are increasingly utilized throughout surgical specialties. These clinical pathways incorporate evidence-based practices to achieve a quicker recovery with fewer complications and shorter LOS [84]. The recommendations span the preoperative, perioperative, and postoperative care periods. While specific guidelines have been created for recovery following PD, the majority of recommendations are the same as for any major abdominal operation [85]. Namely, optimize the patient preoperatively with prehabilitation and nutritional support, perioperatively minimize narcotic use, and post-operatively encourage eternal nutrition, early mobilization, and limit fluid overload.

The recommendation specific to PD is early removal of the perianastomotic drain. We will discuss drain management strategies separately. The 2019 pancreaticoduodenectomy ERAS guidelines also discuss delayed gastric emptying (DGE) but acknowledge there are no effective strategies to prevent DGE. When DGE is suspected, it is important to rule out other underlying causes of poor oral nutritional tolerance, such as an intra-abdominal abscess or POPF [86]. If DGE is confirmed, supplemental nutrition should be considered. What the ideal content of that nutrition is and how it is delivered remains an area of active research. With regard to content, there is emerging data that the addition of “immunonutrients” such as omega-3-fatty acids, glutamine, arginine, and RNA nucleotides, may augment the inflammatory response and thereby decrease postoperative, specifically infectious, complications [86,87]. However, there is insufficient data for the ERAS protocol to recommend immunonutrient feeds. The best route for delivery of nutrition is also debated, while enteral nutrition is generally preferred, a multicentered randomized controlled trial compared nasojejunal early enteral nutrition (NJEEN) to total parenteral nutrition (TPN) following PD and found a higher complication rate in those receiving NJEEN. This included a higher frequency and severity of POPF [88]. Further, while not specifically discussed in the ERAS protocol, in patients with malnutrition, pancreatic exocrine insufficiency should be considered as should the need for pancreatic enzyme replacement therapy. Ultimately, while adherence to ERAS in PD has been variable, adherence is associated with a shorter time to initiation of adjuvant chemotherapy [89,90].

### 4.2. Drain Policies

Surgical drain policies are a crucial aspect of the postoperative course, as the majority of postoperative complications after PD are diagnosed (and often managed) through surgical drains, such as POPF, PPH, and biliary leak. As mentioned previously, early removal of the drains is a central item of the ERAS protocol for pancreatic surgery, with the potential advantage of reducing postoperative major morbidity, specifically B/C POPF [85,91,92,93,94]. Therefore, identifying uncomplicated patients in whom early drain removal is appropriate is crucial. The drain fluid amylase levels (DFA) should be measured on postoperative day (POD) 1 and POD 3. A DFA threshold of three times the upper limit of the institutional normal serum amylase level on or after POD 3 is one of the criteria for current definitions of POPF and biochemical leak (BL) [27]. If such criteria are not met, drains can be removed on POD3. Otherwise, the DFA is reassessed on POD 5 [93,95]. A more dynamic concept for drain management has been proposed and it is based on the DFA slope: if DFA downtrends the risk of BL or POPF appears negligible, irrespective of the initial (POD 1) value of DFA. On the other hand, an increasing value of DFA on POD 3 confers a significant risk for POPF [96]. However, applying these drain management strategies in high-risk patients (FRS 7–10) may be hampered by the presence of external stents. An updated version of this protocol from a high-volume Italian referral center suggests that patients with high-risk pancreas features or the presence of ETS should have the DFA reassessed on POD 5, regardless of POD 1 levels. The presence of an ETS makes previous DFA cut-offs unreliable, and, instead, lower DFA thresholds are used on POD 1 (250 IU/L) to predict BL or POPF [97]. Currently, skepticism remains toward early drain removal protocols based on DFA, despite evidence of improved outcomes, as only 45% of pancreatic surgeons worldwide trust and use DFA levels on POD 1 [81]. Many surgeons still manage surgical drains according to their personal experience and subjective risk assessment, despite existing evidence, and this is probably due to the uncertainty surrounding the outcomes of early drain removal in high-risk patients. A dedicated risk-stratified protocol is still waiting for high-risk scenarios and/or patients with ETS.

### 4.3. Fistula Management

Once a POPF has been diagnosed, it must be managed. There are several indicators of clinical severity, including a high C-reactive protein level and early post-operative hyperamylasemia (POH), by definition, in the first 48 h after surgery [98,99,100,101]. The early recognition and management of POPF’s most severe sequelae (i.e., infected intra-abdominal collections, bleeding, pseudo-aneurysms) before they become clinically relevant is key for improving postoperative outcomes. This can be reached by daily evaluation of standardized parameters and an early, step-up minimally invasive management, as recently demonstrated by a nationwide Dutch, stepped-wedge cluster-randomized trial by Smits and colleagues [102]. In this trial, patients undergoing pancreatic surgery were randomly allocated from usual care to treatment given in accordance with an algorithm for the early recognition and minimally invasive management of postoperative complications. The algorithm determined when to perform abdominal CT, radiological drainage, start antibiotic treatment, and remove abdominal drains, through the standardized daily evaluation (starting on postoperative day 3) of vital signs, abdominal drain output, and serum inflammatory markers (i.e., white blood cell count and C-reactive protein). In the 863 patients in which the algorithm was applied, the primary outcome (a composite of bleeding that required invasive intervention, organ failure, and 90-day mortality) occurred in 8% of patients compared to 14% in the control group, and nationwide 90-day mortality was almost cut in half during the study period.

As mentioned, a step-up approach is usually preferred when managing POPF and related sequelae, going from conservative treatment to antibiotic therapy, and form percutaneous/endoscopic drainage to angiographic treatments. Re-intervention, and, in particular, completion pancreatectomy for POPF, bears a very high mortality rate (higher than 50%) according to a recent systematic review and, in patients in whom a relaparotomy is deemed necessary, a pancreas-preserving procedure seems preferable to completion pancreatectomy [103]. Lastly, in the setting of an ongoing BL/POPF, the nutritional management of patients is also of paramount importance. The best nutrition route provides the correct calory intake, the lower risk of bacterial translocation and sepsis, and does not trigger pancreatic secretion. However, high-level evidence on this topic is lacking [104]. Enteral nutrition appears to be better compared to parenteral nutrition, allowing higher rates and shorter time of POPF closure [105]. The oral route demonstrated similar outcomes if compared to enteral nutrition, ensuring shorter LOS and lower costs [106].

## 5. Conclusions

The choice of surgery for periampullary tumors must be weighed against the risks that PD carries, the alternative therapeutic options, and the multimodality of modern treatments. A comprehensive preoperative risk assessment is key to a tailored clinical recommendation. Once PD is indicated, a sequence of intraoperative and postoperative checkpoints offers clinicians leeway to minimize the burden of surgery.

## Figures and Tables

**Figure 1 cancers-15-02499-f001:**
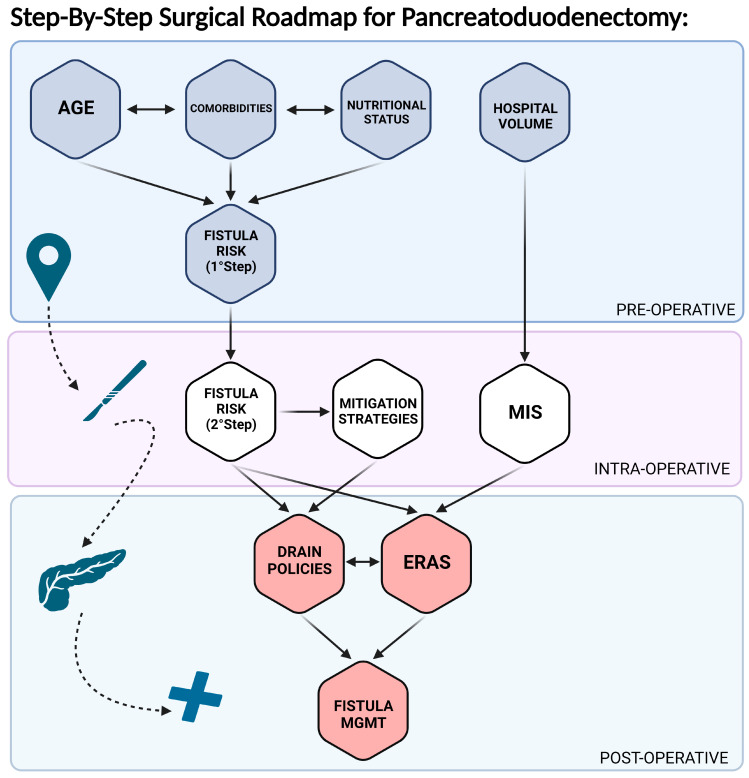
Perioperative roadmap to pancreatoduodenectomy. Arrows indicate possible dependence among items. Abbreviations: MIS, minimally invasive surgery; MGMT, management.

**Table 1 cancers-15-02499-t001:** The modified Frailty Index and the Charlson Age Comorbidity Index for the assessment of risk of patients candidate to pancreatoduodenectomy.

**Modified Frailty Index (mFI)**	**Variable**	**Weight**
	*Functional health status before operation*	
	Totally dependent	1
	*Metabolic*	
	Insulin-dependent DM	1
	*Respiratory*	
	History of severe COPD or current pneumonia	1
	*Cardiovascular*	
	Congestive heart failure within 30d of surgery	1
	MI within 6 months of surgery	1
	Previous PCI, cardiac surgery, or angina within 1 month of surgery	1
	HTN requiring medication	1
	History of revascularization/amputation for PVD, or rest pain/gangrene	1
	*Neurologic*	
	History of TIA	1
	CVA with deficit	1
	Impaired sensorium	1
SCORE: sum of weights INTERPRETATION: score ≥ 3 indicates severe frailty [27,28]		
**Charlson Age Comorbidity Index (CACI)**	**Variable**	**Weight**
	MI Congestive heart failure PVD Cerebrovascular disease Dementia COPD Ulcer disease Mild liver disease DM	1
	Hemiplegia Moderate/severe renal disease DM with end-stage organ damage Leukemia Lymphoma	2
	Moderate/severe liver disease	3
	Metastatic solid tumor AIDS	6
SCORE: sum of weights plus 1 point added for each decade >40 years INTERPRETATION: score ≥ 6 indicates <50% likelihood of being alive 1 year postoperatively [21]		
*DM* diabetes mellitus, *COPD* chronic obstructive pulmonary disease, *MI* myocardial infarction, *PCI* percutaneous coronary intervention, *HTN* hypertension, *PVD* peripheral vascular disease, *TIA* transient ischemic attack, *CVA* cardiovascular accident		

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
