# Peer review of "Current Perioperative Care in Pancreatoduodenectomy: A Step-by-Step Surgical Roadmap from First Visit to Discharge"

_cancers, 2023, doi:10.3390/cancers15092499_

Round 1
Reviewer 1 Report
In the manuscript titled “Current perioperative care in pancreatoduodenectomy: a step-by-step surgical roadmap from the clinic to discharge” Giuliani T et al. present a review on preoperative, intraoperative, and postoperative factors in patients who are planned for or are undergoing a pancreaticoduodenectomy. It‘s a nice and easy readable review, which might be interesting not only for surgeons, but also for oncologist, gastroenterologist and other healthcare specialists. I have few remarks:
1. Some style and writing corrections of the “Simple Summary” are needed.
2. I recommend including a figure with a pathway including the presented factors.
3. Biliary stenting is one of the topics also discussed very often. It would be good to have some data on it also.
-
Author Response
In the manuscript titled “Current perioperative care in pancreatoduodenectomy: a step-by-step surgical roadmap from the clinic to discharge” Giuliani T et al. present a review on preoperative, intraoperative, and postoperative factors in patients who are planned for or are undergoing a pancreaticoduodenectomy. It‘s a nice and easy readable review, which might be interesting not only for surgeons, but also for oncologist, gastroenterologist and other healthcare specialists. I have few remarks:
- Some style and writing corrections of the “Simple Summary” are needed.
R: Thank you for your comments. The Simple Summary has been edited accordingly.
- I recommend including a figure with a pathway including the presented factors.
R: Thank you for your suggestion. We have included a representative figure in the manuscript.
- Biliary stenting is one of the topics also discussed very often. It would be good to have some data on it also.
R: Thank you for thoughtful comment. We have added one session dedicated to the preoperative biliary drainage.
Reviewer 2 Report
In the manuscript entitled "Current perioperative care in pancreatoduodenectomy: 2 a step-by-step surgical roadmap from the clinic to discharge", the authors summerized the roadmap for perioperative care in pancreatoduodenectomy. The manuscript is well organized with clear logic. This content is very helpful for understanding the current status and research progress of periodic care in pancreatodode connectivity. It would be better to provide more in-depth discussions and new perspectives to stimulate readers' thinking. The current form of the manuscript is acceptable in my opinion.
Author Response
In the manuscript entitled "Current perioperative care in pancreatoduodenectomy: 2 a step-by-step surgical roadmap from the clinic to discharge", the authors summerized the roadmap for perioperative care in pancreatoduodenectomy. The manuscript is well organized with clear logic. This content is very helpful for understanding the current status and research progress of periodic care in pancreatodode connectivity. It would be better to provide more in-depth discussions and new perspectives to stimulate readers' thinking. The current form of the manuscript is acceptable in my opinion.
R: thank you for your comment. We hope that our review will provide clinicians dealing with periampullary tumors a practical and up-to-date frame for the clinical decision making process.